# Environment Disaster: A Cross-Sectional Study of the Determinants for the Preparation of Azorean Nurses

**DOI:** 10.3390/healthcare13030303

**Published:** 2025-02-02

**Authors:** Eunice Gatinho Pires, Paulo Nogueira, Maria Adriana Henriques, Miguel Arriaga, Andreia Silva Costa

**Affiliations:** 1Centro de Investigação, Inovação e Desenvolvimento em Enfermagem de Lisboa, Escola Superior de Enfermagem de Lisboa, 1600-190 Lisboa, Portugal; pnogueira@medicina.ulisboa.pt (P.N.); ahenriques@esel.pt (M.A.H.); andreia.costa@esel.pt (A.S.C.); 2Instituto de Saúde Ambiental, Faculdade de Medicina, Universidade de Lisboa, 1649-026 Lisboa, Portugal; miguelarriaga@dgs.min-saude.pt; 3Laboratório Associado TERRA, 1349-017 Lisboa, Portugal

**Keywords:** nursing, climate change, disaster preparedness, disaster management, Azores

## Abstract

Background/Objectives: Climate change increases the vulnerability of regions, communities, and individuals, stressing the urgent requirement to prepare health professionals in alignment with the Sustainable Development Goals. To foster sustainable and resilient communities, it is essential to integrate disaster preparedness into health education and practice. By equipping nurses with essential knowledge and skills, healthcare systems can be better prepared to respond to the challenges of climate change and disasters, contributing to safer and more resilient urban environments. This study aims to identify the factors that determine nurses’ preparedness for disaster situations. Methods: A cross-sectional online survey was conducted through digital platforms among 230 nurses across the Azores to assess their preparedness for disaster management. A structured questionnaire, adapted from the validated Disaster Preparedness Evaluation Tool (DPET©), was administered to registered nurses affiliated with the Nursing Council. The sample was chosen for convenience, and data were analyzed using IBM^®^ SPSS Statistics, employing descriptive statistics, Cronbach’s alpha for measuring internal consistency, independent samples *t*-tests, and one-way ANOVA for comparative analysis. Ethical oversight was obtained from the Ethics Committee of the Azores, ensuring confidentiality and the voluntary nature of participation. Results: The findings revealed concerns about nurses’ disaster response, with 70% of participants indicating low ability. While knowledge was relatively strong (M = 5.50, SD = 1.07), practical competence showed a moderate score (3.51, SD = 1.797). Additional education is necessary in areas such as family preparation (M = 2.58, SD = 1.569), patient management during responses (M = 3.43, SD = 1.312), disaster-specific knowledge (M = 2.95, SD = 1.45), and recovery management (M = 2.53, SD = 1.363). Conclusions: Despite increased knowledge and awareness of climate change and its global impact, there is a need for more meaningful nursing interventions to minimize the impact of climate change on the community. This study highlights that nurses have an in-depth knowledge of communities’ needs, which puts them in a unique position to influence disaster preparation and response. By developing these strategies, nurses contribute significantly to communities’ resilience in climate change, improving society’s ability to respond and adapt to these challenges.

## 1. Introduction

The effects of climate change are evident; the world is threatened by increased natural disasters and extreme weather events, changes in traditional agricultural and fishing areas, loss of biodiversity, and flooding of entire cities, which are increasingly threatening human life, natural systems, and global economies. In Portugal, the Basic Climate Law was drawn up to tackle the problem of climate change, and there are two lines of action: mitigation and adaptation. Mitigation reduces greenhouse gas emissions into the atmosphere, and adaptation minimizes the adverse effects of climate change on biophysical and socioeconomic systems [1]. By preparing the community for adverse events resulting from natural disasters caused by climate change, nurses prevent vulnerability in communities and allow them to adapt to a new reality with low biophysical and socioeconomic impact, emphasizing the importance of their work in preparing the community.

The general climate conditions in the Azores are determined by their geographical location in the context of global atmospheric and oceanic circulation and by the effect of the vast body of water surrounding them. The climate of the Azores is temperate maritime, reflected by the low-temperature range, high rainfall, relative humidity, and persistent winds. These are often affected by the passage of tropical cyclones or tropical storms derived from these and usually result in the worst storms to which the archipelago is subjected [1]. The Azores are vulnerable to adverse natural events, and prioritizing prevention and adaptation interventions is essential to increasing resilience and reducing vulnerabilities.

The International Council of Nursing highlights the importance of disaster nursing skills in providing adequate care and reducing disaster risks [2]. It is common knowledge that nurses play a vital role in direct victim care and disaster management [3,4,5,6,7]. Ziapour et al. [4] report that there are few studies on the perception of nurses’ essential skills in a disaster situation, where it is crucial to investigate the obstacles to preparing nurses for disasters and identify the strategies for preparing nurses for all phases of the disaster management cycle.

In the Azores, nurses must be prepared for disaster situations to develop advanced skills in dealing with people in critical situations, such as triage, infection control, communication, allocating human and material resources, meeting physical and mental health needs, and leadership to ensure adequate response and minimize the impact of the disasters. This study aimed to identify the determinants of nurses’ preparedness for a disaster situation to answer the following questions: (a) What knowledge do nurses in the Azores have about disaster management? (b) How do nurses in the Azores perceive their preparedness for disaster management? (c) What factors determine disaster preparedness?

In 2019, the Government of the Azores developed the Regional Climate Change Programme. One of its strategic objectives is to establish climate scenarios and projections up to 2030 and define program climate change mitigation and adaptation measures. Nurses have a unique opportunity to develop and implement education and preparedness strategies as a form of disaster prevention and a research agenda that responds to current concerns that negatively affect health, including research into the impact of climate change on individuals, families, and communities worldwide.

## 2. Materials and Methods

### 2.1. Study Design

This study presented is a cross-sectional online survey carried out among nurses in the Azores to assess their preparedness for disaster management. The method chosen was a questionnaire, combining structured and open questions, providing an efficient way to collect data quickly. Online dissemination offered nurses flexibility, allowing them to participate at their convenience, while anonymity promoted honest responses.

This study’s scope included all hospitals, both public and private, primary healthcare, and other relevant institutions to ensure complete coverage of the Regional Health Service. The Azores, made up of nine islands with unique geographical and socioeconomic characteristics, face significant challenges in their health system due to their insularity, especially in inter-island travel.

The health structure in the Azores has three public hospitals and one private hospital on the main islands, as well as Island Health Units offering primary and integrated care. Each unit is adapted to the specific needs of its island, ensuring that primary care is widely accessible even with the diversity of specializations on some islands. This comprehensive approach ensures that disaster management policies and practices are effective throughout the archipelago.

### 2.2. Population and Sample

The study sample consisted of 230 nurses from the Azores islands, except for the island of Corvo. The questionnaire was emailed to all nurses registered with the Nursing Council of the Azores and publicized in closed groups on nursing social networks; however, there is no record of participation from the nurses of the island of Corvo. The sample was chosen for convenience, meaning that participants had to be registered with the Nursing Council of Azores, working in the Azores, and available to take part in this study. According to the Nursing Council of Azores, 2424 nurses were registered in December 2023, so this sample represents 10 percent of the population.

The inclusion criteria were broad, accepting nurses with any academic degree (such as a bachelor’s, licentiate, master’s, or doctorate), regardless of gender, age, or length of experience. Voluntary participation was ensured through informed consent, guaranteeing that all participants were aware of the purpose of this study and its implications before agreeing to participate. This approach aimed to gain a diverse and comprehensive insight into nurses’ preparedness for disaster management in the region, reflecting a wide range of experiences and professional qualities.

### 2.3. Instrument

The questionnaire used in this study is based on DPET©, initially developed by Tichy et al. [8]. The continued use of the Disaster Preparedness Evaluation Tool (DPET©), even though it may be seen as outdated by some, has been extensively validated in multiple cultural and professional contexts, as shown in studies in countries such as the United States [8], the Middle East [9,10,11,12], Asia [1,13,14,15,16,17,18], and Portugal [19]. This validation ensures that the instrument is reliable and relevant despite temporal changes. The adapted Portuguese version (DPET-PT) demonstrates how the instrument can be culturally modified to meet the specific requirements of different locations. This flexibility allows the questionnaire to remain relevant in diverse situations and cultures, reflecting local needs. High Cronbach’s alpha coefficients (e.g., 0.94 in the pilot study in Portugal) indicate that the instrument has a strong internal consistency, ensuring that the responses to the items are reliable and valid for measuring the concept of disaster preparedness.

By focusing on competencies related to knowledge and practice in disaster management, DPET© addresses critical elements in disaster preparedness that are inherently linked to the effects of climate change, such as extreme weather events. Disaster knowledge and competencies complement each other but focus on different elements. Knowledge focuses on concepts and information about disasters and includes understanding the types of disaster, their causes and effects, risk factors, historical events, risk assessment, and emergency planning. Competencies refer to practical skills needed to respond to disasters, such as triage, first aid, infection control, communication, and resource allocation. Both are essential for effective preparedness and response, increase confidence for autonomous decisions in unpredictable scenarios, reduce the negative impact of disasters, and improve patient outcomes. The structure of DPET©, which covers various phases of the disaster management process, is versatile enough to be applied in a rapidly changing scenario such as climate change. Despite the realization of possible updates needed, the structure and fundamentals of DPET© remain robust, allowing the tool to be applied effectively in various contexts facing the challenges of climate change.

The survey was conducted through digital platforms, such as institutional emails, the website of the Order of Nurses of the Regional Section of the Azores, and messaging applications, such as WhatsApp and Messenger, over a period of three months, from October to December 2023.

This approach was adopted to maximize participation and ensure that the sample was representative of the target population. The questionnaire used in this study, DPET-PT, is an adapted and validated Portuguese version of the Disaster Preparedness Evaluation Tool (DPET©), developed to evaluate nurses’ preparedness in disaster situations. The Portuguese version was carefully translated and culturally validated to ensure accuracy and relevance in the local context based on the original work [8].

A pilot study was carried out by Santos and Dixe [19] with nurses from mainland Portugal, recording a Cronbach’s alpha coefficient of 0.94, indicating strong internal consistency. The questionnaire, which covers various phases of the disaster management process, consists of 52 questions, with a final scale of 34 items centered on three main factors: post-disaster management, knowledge-related skills (‘knowing’), and practical skills (‘doing’). The fidelity of the DPET-PT was checked by analyzing its internal consistency using Cronbach’s alpha coefficient, which was initially determined using all 45 items, resulting in a total alpha of 0.949. However, 11 items had to be removed because they had Cronbach’s alpha values higher than the overall value and, in some cases, correlation values without the item lower than 0.20. As a result, the DPET-PT scale was left with 34 items.

The nurses in this study perceive their average level of preparedness for disaster situations as low/weak. However, the factor in which nurses are best prepared is competencies related to ‘knowing’. In this pilot study, the training needs most pointed out by the nurses are the role (competencies) of the nurse in a disaster situation and the resources in the community, reference contacts, health departments, emergency contacts, chain of command, and community shelters.

The DPET-PT was applied to nurses in the Azores, allowing their perceptions of disaster preparedness to be assessed. The results highlighted the need for more knowledge in specific areas, indicating where health and educational institutions should invest to improve disaster preparedness and response. The use of the DPET-PT provides potential for practical applications in clinics, research, and training programs for nurses in Portugal.

The first section of the questionnaire consists of 45 items in a six-point Likert scale format, allowing participants to express their degree of agreement, from “strongly disagree” to “strongly agree”. Higher scores reflect perceptions of better preparedness in dealing with disasters. These items are organized into three main dimensions:Post-disaster management: 21 items assessing how nurses perceive their ability to manage activities after a disaster.Competencies: 11 items focusing on the skills needed to act effectively in disaster scenarios.Knowledge: 13 items measuring nurses’ theoretical and practical understanding of disasters.

The questionnaire includes twenty additional open- or closed-response questions in addition to the Likert items. These questions aim to obtain sociodemographic, professional, and academic information and data on training and training needs in disaster management.

The items were grouped into dimensions in the descriptive analysis according to the original DPET. This included variables such as

Knowledge of disasters;Competence in disasters;Preparing the family for disasters;Specific knowledge of disasters;Patient management during disaster response;Knowledge about recovery;Recovery management.

These categories help structure the data analysis, providing a detailed overview of nurses’ preparedness and ability to respond effectively to disaster situations in the Azores.

This study ensured ethical compliance by obtaining the necessary authorization from the Ethics Committee of the Azores, located on the island of São Miguel, and formal consent from the hospital administration to carry it out. At the beginning of the questionnaire, participants were given detailed information about the study’s objectives and the nature of their participation.

It was emphasized that participants had the right to withdraw at any time, with no obligation to justify. The protection of confidentiality and anonymity was strictly ensured, guaranteeing that all information and opinions provided would be treated with the highest level of privacy. These procedures were fundamental to ensuring the trust and freedom of the participants during the data collection process.

### 2.4. Statistical Analysis

IBM^®^ SPSS Statistics (Statistical Package for the Social Sciences, Inc., Chicago, IL, USA) version 26 was used to process and analyze the data. Frequency statistics (absolute and relative) were analyzed for the qualitative variables and descriptive statistics were analyzed for the quantitative variables, using measures of location (mean and median), dispersion (standard deviation), and amplitude (minimum and maximum) [20].

The median represents the value that occupies the central position of the data distribution, i.e., the 50th quartile. Cronbach’s alpha statistics were used to assess the degree of internal consistency between the items that comprise each of the dimensions of the Level of Disaster Preparedness instrument. Cronbach’s alpha values of 0.5 are reasonable. Ideally, they should be between 0.7 and 0.9; the closer they are to one, the greater the internal consistency between them [21].

Students’ *t*-tests for independent samples were used to assess the statistical relationship between dichotomous variables (such as gender, being part of an island with a hospital) and the dimensions of the Level of Disaster Preparedness. The null hypothesis of Student’s *t*-test for independent samples is that the means of the two groups are equal; the alternative hypothesis is that the means of the two groups differ. A significant result (*p*-value < α) means that there is statistical evidence to reject the null hypothesis of equality and accept the alternative hypothesis, concluding that the independent variable (or factor) has a significant effect on the dependent variable (which is being compared) [20]. The parametric Student’s *t*-test requires the data to be normally distributed, an assumption that was assessed using the Shapiro–Wilk test (for groups of fewer than 30 observations) and the Kolmogorov–Smirnov test with Lilliefors correction (for groups of 30 observations or more), under the null hypothesis that the distribution of the data is similar to the normal distribution and the alternative hypothesis that the distribution of the data differs from the normal distribution [20,21].

The normal distribution of the data is an assumption for the application of parametric tests (Student’s *t*-test and ANOVA), which is ensured when there is no statistical evidence to reject the null hypothesis of normality (*p*-value > α). In cases where the assumption of normal distribution of the data was not guaranteed (which occurred in samples/groups larger than 30 observations), it was considered, according to the Central Limit Theorem, that the distribution of the sample averages approaches a normal distribution as the sample size increases. Thus, for sufficiently large samples (generally over 30 observations), the Student’s *t*-test was accepted [20].

One-way ANOVA (Analysis of Variance) was used to compare the mean values of the dimensions of the Level of Disaster Preparedness (dependent variable) between three or more independent samples (groups). The null hypothesis of the ANOVA is that the means are equal against the alternative hypothesis that the means differ. A significant result (*p*-value < α) means that there is statistical evidence to reject the null hypothesis of equality, i.e., at least two groups differ in terms of the mean values of the dependent variable.

In order to identify which groups differ, we used the Bonferroni multiple comparison test, which compares the mean values between the two groups. The null hypothesis of the Bonferroni test is that the means are equal, and the alternative hypothesis is that the means are not equal (i.e., they do not differ significantly) [20].

Spearman’s non-parametric Rho test was used to assess the correlation between years of professional experience and the dimensions of the Level of Disaster Preparedness. Spearman’s Rho correlation coefficient test tests the null hypothesis that the correlation coefficient is zero. The correlation coefficient (r) is a measure that ranges from −1 to +1. The coefficient provides information on the type of association between the variables through the sign: if positive, there is a positive correlation between the variables (high values of one variable correspond to high values of another variable and vice versa); if negative, there is a negative correlation between the variables (high values of one variable correspond to low values of another variable and vice versa); if null (or approximately null), there is no association; and if the coefficient is equal to one, the correlation between the variables is perfect, i.e., there is total mutual influence [20].

In addition to the direction of the association between the two quantitative variables, the absolute value of the coefficient indicates the magnitude or intensity of the association and is considered weak when |r| < 0.25; moderate when 0.25 ≤ |r| < 0.5; strong when 0.5 ≤ |r| < 0.75; and very strong when |r| ≥ 0.75 [20]. The statistical tests used a significance level of 1% (α = 0.01) and 5% (α = 0.05).

## 3. Results

### 3.1. Characterization of the Sample

The characterization of the sample of 230 nurses participating in this study reveals a specific demographic and professional profile (Appendix A): most of the sample is female, representing 75.2% (173 nurses); the predominant age group is between 33 and 43 years old, corresponding to 41.7% (96 nurses); more than half of the participants were married, totaling 56.1% (129 nurses); the vast majority have a bachelor’s degree (88.3%, 203 nurses), while 11.7% (27 nurses) have a master’s degree; 58.3% (134 nurses) work in hospitals; the typical common shift pattern is “rolleman”, practiced by 53.9% (124 nurses); and 30% (33 nurses) have a specialization in Medical-Surgical Nursing. Most work on São Miguel Island (71.3%, 164 nurses) and 91.3% (210 nurses) are on islands with a hospital; 82.6% (190 nurses) have open-ended public service contracts; 81% (187 nurses) work 35 h a week; and the majority, 58.7% (135 nurses), are generalists.

These data provide a detailed understanding of the sample studied, reflecting essential aspects of the sociodemographic profile and the professional context of nurses in the Azores.

### 3.2. Descriptive Analysis of the Disaster Preparedness Level (DPL) of Nurses in the Azores (N = 230)

The descriptive analysis of the mean scores in this study provides clear insight into nurses’ different levels of disaster preparedness, divided into phases of the disaster management cycle. The results highlight areas of strength and those that may require further attention and development (Appendix A):

In the Mitigation and Relief Phase, the average score for patient management during the response was 3.43 (SD = 1.312), indicating a moderate level of preparedness for direct patient management during disasters. Regarding specific knowledge in catastrophe, the average score was 2.95 (SD = 1.45), highlighting an area where further knowledge could be developed.

In the Disaster Preparedness Phase, disaster knowledge achieved the highest mean score of 5.20 (M = 5.50, SD = 1.07), indicating a strong theoretical understanding among nurses regarding disasters. Disaster competence scored 3.51 (SD = 1.797), suggesting an average confidence level in the practical skills necessary to respond to disasters. Family disaster preparedness received an average score of 2.58 (SD = 1.569), highlighting a potential area that requires greater attention and education.

In the Recovery Phase, knowledge about recovery had an average score of 2.93 (SD = 1.411), indicating an understanding that could be enhanced concerning post-disaster recovery processes. Recovery management obtained the lowest mean score of 2.53 (SD = 1.363), suggesting that this area needs better development to support patients and communities after a disaster.

### 3.3. Inferential Analysis (Hypothesis Testing)

This study highlights significant differences in disaster preparedness between male and female nurses across various dimensions, all showing a *p*-value of less than 0.01. Regarding disaster knowledge, men had a higher average score of 3.65 (±0.89) than women, who scored 3.28 (±0.78), indicating that men possess more extensive theoretical knowledge. Similarly, in disaster competence, men scored significantly higher, with an average of 3.22 (±1.09), while women recorded an average of 2.45 (±1.06), suggesting that men feel more competent in their disaster response capabilities. Regarding family preparedness for disasters, men reported an average score of 2.83 (±1.64), whereas women scored 2.30 (±1.40), indicating that men perceive themselves as better prepared to assist their families during such situations. In specific knowledge of disasters, men achieved an average score of 3.13 (±1.31) compared to women’s 2.62 (±1.11), reflecting more excellent detailed knowledge among men. Furthermore, during patient management in response to disasters, men showed an average score of 3.04 (±1.07), while women scored 2.43 (±0.94), indicating that men feel more capable of managing patient care during disaster responses. When examining knowledge about recovery, men scored 3.35 (±1.32) compared to women’s 2.79 (±1.42), signifying a greater understanding of recovery processes among men. However, in recovery management, there were no statistically significant differences between genders, with men averaging 2.41 (±1.18) and women averaging 2.10 (±1.04) with a *p*-value of 0.061.

Overall, the analysis indicates that, except for recovery management, male nurses generally report higher levels of knowledge and competence in disaster preparedness when compared to their female counterparts.

These results indicate specific areas where gender influences perceptions of preparedness and skills in disaster management contexts. This suggests that training interventions could be targeted to address these perceived differences, promoting a balance in preparedness between the genders.

### 3.4. Differences Between Professional Categories

The results indicate that there are no statistically significant differences between the dimensions of the Level of Disaster Preparedness when considering the professional category of the nurses, since the *p*-value is more significant than 0.05 (Appendix A). This suggests that the professional category alone does not significantly influence perceptions or skills related to disaster preparedness among the nurses participating in this study. Thus, the professional category did not prove to be a determining factor in the skills and knowledge assessed. This finding may indicate that, regardless of their position, nurses have similar levels of preparedness, possibly due to standard training or professional practice factors.

### 3.5. Differences Between Professional Roles 

This study revealed significant differences in specific dimensions of disaster preparedness related to the nurses’ professional roles. Nurses in managerial or coordinating positions demonstrated higher scores in disaster knowledge, with Nurses in Charge averaging 3.93 (±0.98) and Nurses in a Coordination Role scoring 4.15 (±0.95), both outperforming Generalist Nurses, who had an average of 3.21 (±0.70), and Specialists, who scored 3.45 (±0.88), with the coordination role particularly standing out. Regarding competence in disaster management, nurses in a coordinating role achieved significantly higher scores, averaging 3.41 (±1.41), compared to general nurses, who averaged 2.53 (±1.03). Furthermore, nurses in a coordinating role scored 3.85 (±1.63) in recovery knowledge, showing notably more excellent knowledge than specialist nurses, who had an average score of 2.66 (±1.41) (Appendix A).

For the other dimensions analyzed, such as preparing the family for the disaster, specific disaster knowledge, patient management during the response, and management during recovery, there were no significant differences related to professional function, with a *p*-value >0.05. This suggests that, for these areas, the role played in the work environment is not a determining factor in nurses’ disaster preparedness.

These results indicate that leadership and coordination roles may be associated with a more excellent perception of preparedness and competence in some specific areas, possibly due to expanded responsibilities or more significant involvement in disaster management planning and training.

### 3.6. Differences Between the Islands 

The island location of nurses in the Azores significantly influences their perception of disaster preparedness, with notable variations observed between the islands. Regarding disaster knowledge, nurses from the island of Santa Maria achieved significantly higher scores, averaging 4.53 (±0.88), compared to those from São Miguel, who averaged 3.23 (±0.73). In disaster competence, Santa Maria again stood out with higher scores of 4.12 (±0.55), while nurses from São Miguel scored only 2.44 (±1.02). When it comes to preparing the family for disaster, nurses in São Jorge reported a substantially higher average of 5.33 (±1.15) compared to their counterparts in São Miguel, who scored 2.20 (±1.37). In specific knowledge of disasters, Santa Maria also performed better with an average score of 4.44 (±1.19) compared to São Miguel, which scored 2.54 (±1.08). Finally, in recovery management, Santa Maria scored higher at 3.87 (±1.04) compared to Faial, which scored 2.10 (±0.96), and São Miguel, which had an average of 2.00 (±0.96). Terceira also outperformed São Miguel with a score of 2.70 (±1.34). On the other hand, no statistically significant differences were observed in the dimensions of patient management during the response and knowledge about recovery, suggesting that these aspects of disaster preparedness are more uniform across the islands (Figure 1 and Figure 2).

These differences indicate that the island of residence influences nurses’ Level of Disaster Preparedness. São Jorge’s recent seismic-volcanic crisis may contribute to enhanced knowledge and resilience among professionals there. Factors such as access to resources, specific training, and previous experience may contribute to these variations, highlighting the importance of adapting training and support strategies to each island’s particularities.

Comparison of NPC dimensions between islands—one-factor ANOVA test.

### 3.7. Differences Between Islands with and Without a Hospital 

The results indicate statistically significant differences between nurses working on islands without a hospital compared to those working on islands with a hospital, with the former recording higher scores in all dimensions of disaster preparedness.

In terms of knowledge of disasters, nurses on islands without hospitals had an average score of 4.02 (±0.91), compared to 3.31 (±0.79) for those on islands with hospitals. The t-value was 3.771 and the *p*-value was 0.000, indicating a statistically significant difference. For disaster competence, the average score for nurses on non-hospital islands was 3.43 (±0.92), whereas those on hospital islands scored 2.57 (±1.10). The t-value here was 3.390, with a *p*-value of 0.001, reflecting a significant difference. When preparing families for disaster, nurses in non-hospital settings averaged 3.30 (±1.94), while those in hospital settings scored 2.35 (±1.40). The t-value was 2.147, and the *p*-value of 0.044 suggests a significant difference. In specific knowledge of catastrophes, nurses without hospitals had an average score of 3.77 (±1.25), compared to 2.65 (±1.13) for those with hospitals. This resulted in a t-value of 4.174 and a *p*-value of 0.000, indicating strong statistical significance. For patient management during the response, non-hospital nurses averaged 3.21 (±0.94), whereas hospital-based nurses scored 2.52 (±1.00). This yielded a t-value of 2.969 and a *p*-value of 0.003, demonstrating significance. In knowledge about recovery, nurses in non-hospital settings had an average of 3.80 (±1.28) compared to 2.84 (±1.40) for those in hospitals, with a t-value of 2.948 and a *p*-value of 0.004, indicating a significant difference. Finally, regarding recovery management, nurses on non-hospital islands averaged 2.86 (±1.19), while those on hospital islands scored 2.12 (±1.05). The t-value was 2.983, and the *p*-value of 0.003 shows a statistically significant difference between the two groups (Table 1).

This suggests that nurses on islands without hospitals may develop a broader or deeper set of skills and knowledge due to the need to deal with a wider range of situations without the hospital’s specialized approach. Such a context may require greater autonomy and more comprehensive preparation, as specialized resources and support may be less accessible.

These findings highlight the importance of understanding how different working environments and access to resources influence disaster preparedness. They suggest that training and support need to be tailored to meet the specific needs of nurses in different contexts.

### 3.8. Factors That Can Contribute to the Difference Between Islands with a Hospital and Islands Without a Hospital

Developing Resilience: The lack of a hospital in the community can promote greater awareness of the need to prepare themselves, their families, and the community to face disasters. Nurses working in environments without hospitals tend to develop high levels of resilience, creating adaptability and the ability to self-manage in challenging situations. The research by [22] highlights that nurses can develop resilience to cope with clinical pressures and demands in extreme conditions through emotional adaptation and innovation in care.Innovative Capacities: The absence of hospitals forces nurses to develop innovative practices in crisis care. They need to deal with the situation creatively and effectively to respond to the needs of the victim, family, and community [22].Importance of Planning: The literature emphasizes that planning and preparation must occur before disasters occur. This study highlights that more than half of the nurses know their unit’s disaster plan, associated with a better perception of preparedness. Awareness of the protocols and participation in training programs on disasters are essential factors that contribute to the adequate preparation of nurses [23].

Although disasters cannot be avoided, preparation is crucial. Healthcare institutions and nurses should continuously focus on preparing before, during, and after disaster events. This includes reading and understanding disaster plans and engaging in specific educational modules that address post-disaster interventions and care.

### 3.9. Differences Between Specialties (Nursing Specialization Area) 

This study highlighted significant differences in perceptions of disaster preparedness among nurses from various specialties (Appendix A). In terms of disaster knowledge, Maternal and Obstetric Health Nurses had the lowest values, averaging 2.66 (±0.36), compared to Community Nursing, which averaged 3.74 (±0.97), and Medical-Surgical Nursing, with an average of 3.77 (±0.87). This suggests that Maternal and Obstetric Health Nurses may have less confidence or experience in disaster scenarios.

For disaster competence, scores for Maternal and Obstetric Health Nurses were significantly lower, averaging 1.61 (±0.53), compared to Medical-Surgical Nurses, who scored 3.15 (±1.18). In recovery knowledge, Mental Health and Psychiatric Nurses reported significantly higher values, averaging 4.22 (±1.56) versus 1.86 (±1.21) for Maternal and Obstetric Health Nurses.

Management in recovery was notably higher among Mental Health and Psychiatric Nurses, who had an average score of 3.44 (±1.36), compared to those in Community Nursing, Child Health, and Pediatrics, both averaging 2.08 (±1.01 and ±0.90, respectively), and Maternal Health and Obstetrics, which had a low average of 1.37 (±0.48).

Other dimensions, such as family preparation for the disaster, specific disaster knowledge, and patient management during the disaster response, showed no significant differences between specialties, with a *p*-value > 0.05.

These results suggest that areas of specialization significantly impact perceived preparedness in certain aspects of disaster management. Specialties more involved with critical response situations, such as Medical-Surgical Nursing, or with comprehensive approaches, such as Mental Health and Psychiatry, appear to be more prepared to deal with the complexities of disaster. This points to areas where training and development could be focused for other specialties, especially Maternal and Obstetric Health.

In addition, nurses who attended training courses focused on disaster management demonstrated higher competencies. This emphasizes the importance of formal education and specialized training to improve preparedness in disaster response [4].

### 3.10. The Role of Experience and Training

The results also suggest that nurses in intensive care units and those with more training and experience tend to perform better in preparedness. Exposure to critical scenarios can lead to greater confidence and ability to respond in disaster situations, highlighting the need for comprehensive training and education programs for all nurses, given their crucial role in disaster response [12].

### 3.11. Effect of Length of Experience 

The results indicate a positive and significant correlation between the nurses’ length of professional experience and the family’s level of preparedness for the disaster. Specifically, the correlation of the level of family preparedness with the total time of professional experience is (r = 0.146) and that with the experience in the current unit is (r = 0.133), both significant at the 5% level (*p* < 0.05) (Table 2) (Figure 3).

Despite being weak correlations, they suggest that the more years of professional experience a nurse has, the greater their family’s perceived preparedness to deal with disasters tends to be. This can be attributed to more experienced nurses accumulating more knowledge and skills over the years, which they can also pass on to their families. These results highlight the importance of continuous practical experience not only for the professional development of nurses but also for preparing their families for disaster situations.

The data presented a comprehensive overview of respondent’s preparedness and experiences regarding disaster management in a healthcare context.

Preparedness and Training Needs (Appendix A): A substantial majority (74.6%) of respondents identify training as the primary area for improvement to be better prepared for disaster situations. This underlines the critical need for formal education and skill development among healthcare professionals in handling disasters. While simulations and drills are essential, they account for a smaller percentage (13% solo and 7.6% in combination with training). This indicates an opportunity to enhance engagement in practical, hands-on training, which complements theoretical knowledge.

Priority Level for Preparedness (Appendix A): Approximately 79.7% of participants view disaster preparedness as a maximum priority. This signifies a shared recognition of the importance of being ready to respond effectively to disaster, hinting at an organizational or systemic interest in improving disaster management training.

Current Capabilities in Disaster Response (Appendix A): A significant 68.8% of respondents rated their capacity to respond to disasters as low. This raises concerns about confidence and readiness in crisis situations, suggesting urgent investments in training and resources to improve proficiency.

Experience with Training and Simulations (Appendix A): While 58.3% participated in simulations as nurses, only a small fraction (18.3% each) received disaster-related education in their undergraduate and postgraduate nursing programs. This highlights a potential gap in formal education regarding disaster preparedness within nursing curricula. A strong desire for additional education is evident, especially concerning roles and competencies during disasters (83.5%) and knowledge of biological and chemical agents (80.9%). This suggests that respondents are eager to enhance their skills and contribute effectively to disasters.

Regularity of Drills and Their Effectiveness (Appendix A): A significant portion (71.3%) of respondents reported that their workplaces do not conduct regular disasters drills. This lack may contribute to the low self-assessment of preparedness and highlights a critical gap in practice. Among those participating in regular drills, a majority (65.6%) felt that these efforts were practical. However, there was a variation in perception among those who reported no regular simulation exercises, indicating a potential disparity in understanding what constitutes practical preparedness training.

Authentic Experience in Disaster Situations (Appendix A): An overwhelming 85.7% of respondents have not participated in a real disaster situation. This lack of experience can contribute to the low confidence levels reported, reinforcing the need for more exposure and training in realistic scenarios. The data presented insights into healthcare professionals’ experiences during a disaster, explicitly reflecting on the 2021 seismic crisis in São Jorge and general preparedness and capacity to respond to emergencies.

Disaster Experiences (Appendix A): The responses capture various disaster scenarios, with a notable emphasis on seismic events such as the crises in 2021 and 2022 in São Jorge and important historical earthquakes and their aftermath. Other incidents include transportation accidents with multiple casualties, floods, landslides, and the ongoing challenges posed by the COVID-19 pandemic, indicating a wide range of crisis management experiences faced by healthcare professionals.

Specific Experiences during the Seismic Crisis: One in-depth account described the chaos of the earthquake on 19 March 2021. The respondent detailed their actions of remaining calm, ensuring the patient’s oxygen supply, and coordinating patient transfers. This underscores the essential role of nurses in managing patient care during disasters amidst significant disruption. The respondent also highlighted the importance of being prepared with emergency supplies (such as compresses, bandages, saline, and medications), illustrating crucial crisis preparedness and response elements in the healthcare setting.

Impact of Continuous Support: The statement regarding ongoing support for elderly patients who had to leave their homes due to the earthquake reflects the ongoing impact of disasters and the necessity of long-term care and recovery strategies in institutional settings.

Preparedness to Respond to Disasters (Appendix A): The data indicate that respondents’ feelings of preparedness vary: 42.6% felt prepared, while 38.3% did not feel prepared at all, and 19.1% indicated a moderate level of preparedness but acknowledged the need for more training. The relatively high percentage of respondents feeling unprepared or moderately prepared suggests needs in training or experience, which may impact response efficacy during crises.

The responses imply a significant need for improved training programs, as well as education, focused on disaster response protocols, triage, and crisis management skills. Given that many professionals feel unprepared, targeted training could enhance confidence levels and operational readiness during future disasters.

The perceived preparedness may be linked to actual experience in disaster scenarios. Those with more extensive training or prior firsthand experience may feel more equipped to handle crises, while those lacking such experiences might feel overwhelmed.

## 4. Discussion

This study’s results highlight alarming concerns regarding nurses’ preparedness to respond to disasters. The world faces growing threats due to increased natural disasters, extreme meteorological phenomena, changes in agricultural and fishing areas, loss of biodiversity, and flooding of cities and even entire countries. The fight against climate change requires collective action and a coordinated response from the international community [24]. In the context of the climate agenda, the involvement of nurses in the defense of causes and policies for the benefit of public health is essential. Nurses’ leadership is crucial in tackling health and social crises, and it is vital to understand the structural and social determinants of communities [25]. Additionally, equipping nurses with the skills to manage disaster situations—emphasizing prevention and well-being—is essential for reducing vulnerability to climate risks, enhancing resilience to extreme weather events, and improving emergency response capabilities. In this study, 70% of participants reported a low capacity to respond, highlighting an urgent need to address this shortfall. These findings align with previous surveys indicating a global inadequacy in nurses’ disaster preparedness [7,8,23,26,27].

### 4.1. Determinants of Disaster Preparedness

The adverse effects of climate change represent a significant challenge for humanity worldwide, contributing to the global disease and premature mortality burden [28]. Understanding the impact of climate change, mainly through the lens of natural disasters and effective adaptation strategies, is crucial for enhancing resilience against its harmful consequences.

Previous experience is crucial for knowing and preparing nurses for disaster response. This study revealed that only 14.3% of nurses had engaged in an absolute disaster; of these, only 42.6% felt prepared. Active involvement in disasters boosts confidence and allows nurses to apply theoretical knowledge practically. Previous experience makes it possible to identify a community’s vulnerabilities and consequences. Hossain et al. [29], in their study, identified hunger (83%) and physical injuries (56.8%) as the most significant complications of climate changes. Difficulties in obtaining first-aid medicines to deal with sudden health problems or access to health institutions that have been closed due to the storms contribute to the community’s increased vulnerability. Increased exposure and training lead to improved readiness, as supported by Ziapour et al. [4], who found that emergency nurses with both experience and training scored higher in preparedness. Nurses with experience and specific knowledge of the adverse effects of climate change can contribute positively by identifying adaptation strategies to increase access to food quickly and adequately, reduce traumatic events, or adapt resources for an adequate response.

Previous disaster situations enhance the understanding of roles and expected behavior. Practical knowledge in real scenarios is often more valuable than years of experience alone. While nurses understand triage, comprehensive training and disaster management simulations are needed. Mastering first aid and triage is crucial for nurses to provide immediate care and lifesaving interventions in disasters, as noted by Santos and Rabiais [5].

Nurses often serve as first responders in hospitals and communities during disasters, and their practical experience directly influences their sense of competence and preparedness [15]. Providing nurses with specific disaster-related skills enhances their autonomy, allowing them to make informed decisions and manage situations independently [6]. While nurses have a moderate understanding of Personal Protective Equipment (PEE) use in bioterrorism, they need better knowledge of decontamination procedures. Improving nurses’ knowledge of PPE types, limitations, maintenance, and proper usage is crucial to prevent cross-contamination [5].

Hassan et al. [30] emphasize the need for health professionals to be more aware of climate change and its negative impact on a community’s health. Active involvement in disaster enhances nurses’ clinical judgment and preparedness for disaster events, as practical experience develops skills beyond theoretical training. This study highlights a critical need for better disaster preparedness among nurses. Barriers to effective response include a lack of family emergency plans, impacting nurses’ focus on disaster management. Tichy et al. [8] stress that nurses must ensure their families’ safety to perform effectively. However, this raises ethical dilemmas as nurses navigate their professional responsibilities while protecting loved ones, a challenge noted by Brewer et al. [28]. This situation can be circumvented if family members and neighbors support one another when faced with these problems [29].

Understanding the public’s vulnerability to the health impacts of climate change is crucial for timely preparation and early prevention [29]. Azorean nurses are aware of potential disaster risks and acknowledge their limitations in knowledge and competence. However, they feel inadequately prepared to manage these situations and express a strong need for additional training. Effective planning addresses disasters, mitigates risks, ensures safety, and maintains operational continuity. Low literacy and low wealth levels are associated with greater vulnerability [29,30]. Health institutions should create proactive risk management plans that involve healthcare professionals in their development. Engaging nurses in disaster planning enhances risk awareness and promotes early readiness, contributing to positive outcomes.

Nurses who are involved in drawing up and implementing disaster plans are essential for dealing with health and social crises [25,29,30,31,32]. This study indicates that nurses are largely unfamiliar with organizational logistics in disaster response due to infrequent drills (71.3%) and limited participation in emergency planning. This disengagement hampers their understanding of institutional and individual roles during disasters. Active involvement in response planning is vital as it enhances nurses’ knowledge and effectiveness, leading to improved preparedness perceptions and reduced uncertainties [12]. Health institutions should establish comprehensive knowledge of the health impacts of climate change for disaster plans and provide sufficient training to ensure effective interventions during and after disasters. Currently, nurses have minimal involvement in developing emergency plans, as noted in previous research [8]. Their engagement in these processes can enhance their effectiveness in emergencies. Familiarity with disaster plans is linked to better preparedness and greater involvement in health policymaking and compliance, thus improving nursing practice quality [5]. Understanding response phases, roles, and the chain of command in hospital emergency plans is essential for enhancing disaster response [5,30].

The combination of experience in disaster response and previous training contributes positively to the perception of preparedness. The results of this study align with Ziapour et al. [4], who, in their study, point out that emergency nurses with both experience and training in disasters scored significantly higher in terms of preparedness than those who do not have this combination. Education should be prioritized to recognize the signs and symptoms of climate change-related diseases, and competencies in adaptation, mitigation, and resource mobilization should be improved at all levels [30]. This highlights the importance of continuous training programs and practical exercises that increase the relevance of training and build nurses’ confidence in their ability to manage disaster situations.

One significant impact of climate change is the 40% increase in emergency room visits due to trauma (adult and pediatric), emergencies due to carbon monoxide poisoning, skin infections, respiratory diseases, medical illnesses, and psychotic outbreaks due to mental health decompensation [31], which means that specialized nurses are needed for the different age groups, as well as knowledge of the different areas of activity. This study reveals the urgent need for continuous and practical training for nurses who deal with disaster situations. Most nurses in this study (80%) prioritize disaster prevention and management in their curricular and professional education. This is in line with Ziapour et al. [4], whose study involved nurses who work in emergency services in Turkey, face floods and earthquakes frequently, and consider that they have the necessary technical skills due to their awareness of the need for training but have difficulty in their clinical judgment and in identifying diagnoses. Promoting a culture of preparedness is essential because a significant number of nurses (74.6%, n = 138) identified the need for training to feel better prepared for disaster situations, with 13% (n = 24) pointing to the importance of practical simulations. Creating a culture of preparedness in healthcare organizations is vital to strengthening response capabilities. The knowledge gained should be used to inform resource allocation and strengthen preparedness and response in healthcare systems [12].

While nurses understand the screening system, there is a strong need for enhanced training programs. Although annual drills by the local airport management aid in knowledge acquisition, regular training is still required. Many nurses lack opportunities to participate in simulations, indicating a gap in practical training and exposure to real disaster scenarios [4,12,14]. Given the importance of communication during disasters, training should specifically address communication skills and involve the media as a means of raising awareness and disseminating information to the community [30]. Implementing practical and interactive workshops focused on disaster scenarios will enable nurses to practice and develop essential skills. Investing in communication-oriented training can significantly improve effectiveness and coordination during disaster responses, resulting in better patient care and crisis management outcomes [4,30].

Given the specific nature of disasters that can affect the region (such as earthquakes and floods in the Azores), the region’s isolation from the mainland is a reality, which can lead to a reduction in food supplies, technological resources that are unable to function, and difficulty in accessing healthcare, placing the Azores in a situation of increased vulnerability. Training must be tailored to address the unique risks of the local area, and the community’s needs and disaster management skills should be incorporated into the health institution’s ongoing training, with particular attention paid to less experienced nurses [33]. This training should be aligned with international standards and adapted to each department’s needs. Nurse managers should periodically evaluate nurses’ competencies to ensure that training remains up-to-date and effective [34]. Nurses expressed concerns about a need for more training and familiarity with disaster protocols, which makes it vital that health administrators use the results of these surveys to establish health standards and protocols at a national level [4,24,30,35]. Training should include face-to-face activities, such as simulations and crisis response exercises, which can improve the effectiveness of disaster management education [33].

Disaster nurses are defined as nurses with the expertise to help victims by minimizing life-threatening health risks; we can assume that nurses are expected to adapt to their environment and disaster conditions [7]. Adaptation strategies are important to increase a community’s resilience to the health impacts of climate change [29]. Training programs must focus on technical knowledge and strengthen nurses’ critical thinking and communication skills. This is vital if they are to be able to assess disaster situations and make informed decisions quickly, requiring competency in reporting symptoms or events that might indicate the onset of an emergency and maintaining ongoing assessment of patients/families/communities for needed changes in care during evolving disaster events [5].

The need for leadership and collaboration at regional, national, and international levels is important for smaller countries with limited resources to combine efforts and be more effective and efficient in disaster management. The disruption of access to healthcare due to damage to medical facilities, power, transport, communications, or other infrastructure due to extreme weather-related disasters can negatively affect people [30]. This study recognizes leadership as a crucial factor in effective disaster management. In disaster situations, leading teams and making quick, decisive decisions are vital [14]. Not only do nurses play a key role in providing care, but they also often take on leadership roles during catastrophic events. It is imperative that training programs include components focused on developing leadership skills. This would help nurses prepare to lead their teams during crises, facilitating a more effective and coordinated response. Health professionals must be prepared to intervene during and after disasters. Knowledge of the unit’s disaster plans is associated with a better perception of preparedness, and familiarity with these plans can eliminate uncertainty and positively impact the response [12].

Learning to lead under pressure involves developing the ability to make quick and appropriate choices, which can be crucial to meeting patients’ needs and the team’s safety. Nurses are critical throughout the disaster response process, providing care to minimize health risks and life-threatening damage that can occur during and after a disaster. Their leadership skills and competence enable them to act effectively in a chaotic and rapidly changing environment. Competency in using specific disaster plans, including the chain of command, is crucial for effective incident management during events, exercises, or drills. Developing cognitive flexibility to inhibit non-functional responses and make better decisions in complex situations is essential for effective disaster response [5]. Investing in communication training is also crucial to ensuring that information is transmitted clearly between teams, improving coordination in disaster response. Enhancing leadership skills, motivation, and willingness to face adverse occurrences is necessary for managing disaster situations effectively [5].

The diagram (Figure 4) highlights the importance of knowledge and leadership in enhancing disaster preparedness among nurses. Adequate preparation involves cultivating a deep understanding of the negative impacts of climate change, with a preventive and adaptative perspective fostering relevant training and strengthening decision-making skills within leadership roles. This holistic approach can significantly improve nursing responses to disasters, ultimately enhancing patient care and safety during disasters.

### 4.2. Disaster Advanced Nursing

The literature considers Advanced Nursing Practice as the specialized knowledge a qualified professional nurse uses to make complex and advanced decisions and put clinical skills into practice by integrating theory, practice, teaching, research, leadership, and management [36]. The study results show that most nurses (84%) need more training related to their role and the skills required in disaster situations. This reflects nurses’ awareness of the complexity of their responsibilities in disaster contexts and the importance of preparing for an adequate response.

The practice of Advanced Nursing implies that nurses must continually educate themselves on the best practices and technologies available in the provision of healthcare during and after disasters. Participation in prevention and mitigation activities increasingly goes beyond the traditional concept of providing care, involving a proactive stance towards preparedness. It involves the participation of nurses in the development of health policies and management programs, which contributes to a more coordinated and effective approach throughout the disaster cycle.

### 4.3. Implications and Limitations of This Study

This study reveals a valuable opportunity to improve nurses’ disaster preparedness, contributing to the Sustainable Development Goals, and to reduce the negative impacts of climate change, especially in strengthening resilience and adaptive capacity to disaster-related risks. Improving this preparedness not only raises the quality of healthcare but also supports the building of safer and more sustainable communities, aligning with global efforts to reduce the risk of catastrophe and its negative impact and promoting a more stable and secure environment.

This study highlights some considerations that should be made about the results. Although it focused on nurses from the Azores, it provides valuable insight into local practices, reflecting the unique characteristics of this region. Such a focus allows for a deeper analysis of the reality faced by health professionals in the Azores, although it may limit the generalizability of the results to other regions.

Collecting data through questionnaires has helped gather information efficiently, although it may introduce some variability in the responses. This study’s cross-sectional nature, in turn, provides a valuable snapshot of disaster management skills, even if it does not allow us to observe changes over time. Understanding these nuances is essential to interpret the results critically and guide future research. This reflection could enrich nursing research and promote the continuous development of the skills needed for disaster management.

## 5. Conclusions

This study underscores the critical need for enhanced disaster preparedness among nurses, particularly in disaster-prone regions such as the Azores. With the increasing impacts of climate change, it is urgent to recognize the importance of disaster training and implement essential procedures to ensure effective responses. Continuous education and training programs that seamlessly integrate theory and practice in disaster scenarios are vital for strengthening nurses’ capacity to act decisively. Training should encompass family safety planning, which can help reduce anxiety and increase focus during disasters.

Moreover, it is essential to tailor training to regional needs, especially in areas lacking hospital resources, to promote ongoing knowledge regarding disaster response plans and the specific challenges posed by climate change.

Additionally, healthcare institutions should prioritize the development of leadership skills among nurses, enabling them to guide their teams effectively during crises.

This proactive approach will enable nurses to fulfill pivotal roles in disaster response and community resilience, ultimately benefiting both healthcare professionals and the communities they serve in times of crisis exacerbated by climate change. By fostering leadership, improving knowledge, and enhancing community resilience, we can create a more robust healthcare system prepared to face future challenges.

## Figures and Tables

**Figure 1 healthcare-13-00303-f001:**
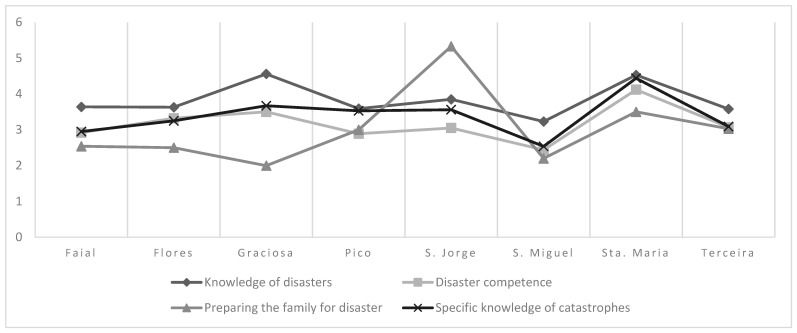
Average values of the dimensions by island (Part 1). Note: ‘Preparing the family for disaster’ is significantly higher in São Jorge than in São Miguel. The other dimensions are significantly higher in Sta. Maria than in S. Miguel.

**Figure 2 healthcare-13-00303-f002:**
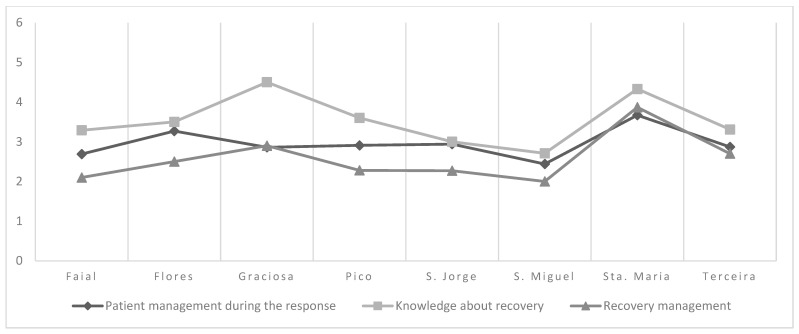
Average values of the dimensions per island (Part 2). Note: ‘Recovery management’ is significantly higher in Sta. Maria than in Faial and S. Miguel, and in Terceira than in S. Miguel. The other dimensions do not differ significantly.

**Figure 3 healthcare-13-00303-f003:**
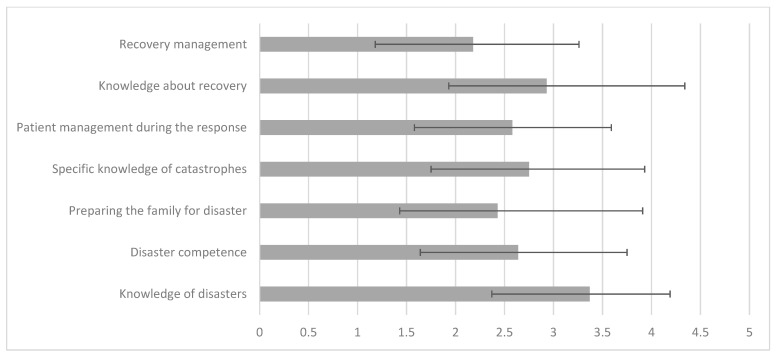
Mean values of the dimensions and their standard deviation in the sample (n = 229). Note: ‘Preparing the family for disaster’ has a positive and significant correlation, at the 5 per cent level, with ‘Years of professional experience’ (r = 0.146) and with ‘Years of professional experience in the current unit’ (r = 0.133). The other dimensions are not significantly influenced by length of professional experience.

**Figure 4 healthcare-13-00303-f004:**
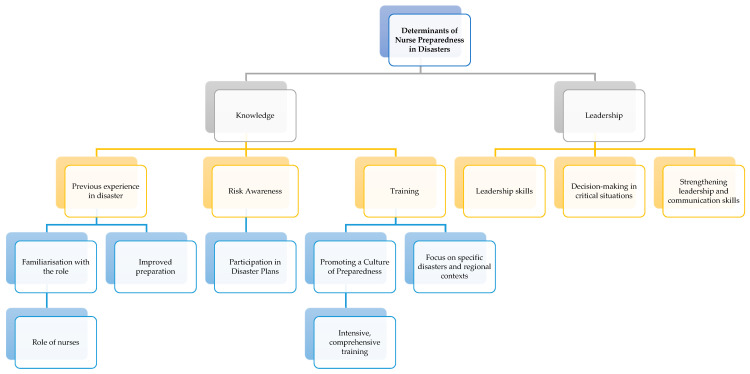
Determinants of nurse disaster preparedness.

**Table 1 healthcare-13-00303-t001:** Comparison of NPC dimensions between island nurses with and without a hospital—Student’s *t*-test for independent samples.

	Island with Hospital	n	Average	Standard Deviation	*t* (gl) *p*-Value
Knowledge of disasters	No	20	**4.02**	0.91	*t* (228) = 3.771 *p*-value = 0.000
	Yes	210	3.31	0.79	
Disaster competence	No	20	**3.43**	0.92	*t* (228) = 3.390 *p*-value = 0.001
	Yes	210	2.57	1.10	
Preparing the family for disaster	No	20	**3.30**	1.94	*t* (20,95) = 2.147 *p*-value = 0.044
	Yes	210	2.35	1.40	
Specific knowledge of catastrophes	No	20	**3.77**	1.25	*t* (228) = 4.174 *p*-value = 0.000
	Yes	210	2.65	1.13	
Patient management during the response	No	20	**3.21**	0.94	*t* (228) = 2.969 *p*-value = 0.003
	Yes	210	2.52	1.00	
Knowledge about recovery	No	20	**3.80**	1.28	*t* (228) = 2.948 *p*-value = 0.004
	Yes	210	2.84	1.40	
Recovery management	No	20	**2.86**	1.19	*t* (228) = 2.983 *p*-value = 0.003
	Yes	210	2.12	1.05	

Legend: n—number of valid cases (absolute frequency); *t*—Student’s *t*-test for independent samples; gl—degrees of freedom; *p*-value—probability of significance.

**Table 2 healthcare-13-00303-t002:** Spearman Rho correlation coefficient test.

	Average	DP	Years of Professional Experience	Years of Professional Experience in the Current Unit
Knowledge of disasters	3.37	0.82	0.076	0.085
Disaster competence	2.64	1.11	−0.043	−0.017
Preparing the family for disaster	2.43	1.48	0.146 *	0.133 *
Specific knowledge of catastrophes	2.75	1.18	−0.082	−0.042
Patient management during the response	2.58	1.01	−0.036	0.020
Knowledge about recovery	2.93	1.41	−0.040	−0.011
Recovery management	2.18	1.08	−0.007	0.005

* Significant at the 5% level.

## Data Availability

The data presented in this study are available on request from the corresponding author.

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
