# Peer review of "Environment Disaster: A Cross-Sectional Study of the Determinants for the Preparation of Azorean Nurses"

_healthcare, 2025, doi:10.3390/healthcare13030303_

Round 1
Reviewer 1 Report
Comments and Suggestions for Authors
Interested topic and well written.
Please elaborate more about sampling and how did you identify the representative sample size.
Please explain the results of the pilot and was their any modifications on the tool
Please explain why did you use this tool about dated disaster preparedness tool and how it covers the areas of climate or environment disasters.
Please rewrite the the results in paragraphs and avoid the dot points. You may provide little explanations for tables including demographics, findings and inferential analysis.
Please rewrite the the the discussion in paragraphs instead of dot points and critically discuss the main findings with other other literature of climate change.
Author Response
Comments 1: Please elaborate more about sampling and how did you identify the representative sample size
Response 1: Agree. We explain and modified the sentence to clarification. Line 106-113
“The sample was chosen for convenience, meaning that participants had to be registered with the Nursing Council of the Azores, working in the Azores, and available to take part in the study. According to the Nursing Council, 2,414 nurses were registered in December 2023, so this sample represents 10 percent of the population.”
Comments 2: Please explain why you used this tool about dated disaster preparedness tool and how it covers the areas of climate or environment disasters.
Response 2: Agree. We explain and modified the sentence to clarification. Line 124-142
“The continued use of the Disaster Preparedness Evaluation Tool (DPET©), even though it may be seen as outdated by some, the DPET© has been extensively validated in multiple cultural and professional contexts, as shown in studies in countries such as the United States, the Middle East, Asia, and Portugal. This validation ensures that the instrument is reliable and relevant despite temporal changes. The adapted Portuguese version (DPET-PT) demonstrates how the instrument can be culturally modified to meet the specific requirements of different locations. This flexibility allows the questionnaire to remain relevant in diverse situations and cultures, reflecting local needs. High Cronbach's alpha coefficients (e.g., 0.94 in the pilot study in Portugal) indicate that the instrument has a strong internal consistency, ensuring that the responses to the items are reliable and valid for measuring the concept of disaster preparedness. By focusing on competencies related to knowledge and practice in disaster management, DPET© addresses critical elements in disaster preparedness that are inherently linked to the effects of climate change, such as extreme weather events. The structure of DPET©, which covers various phases of the disaster management process, is versatile enough to be applied in a rapidly changing scenario such as climate change. Despite the realization of possible updates needed, the structure and fundamentals of DPET© remain robust, allowing the tool to be applied effectively in various contexts facing the challenges of climate change.”
Comments 3: Please explain the results of the pilot and was there any modifications on the tool
Response 3: Agree. We explain and modified the sentence to clarification. Line 157-170
“The questionnaire, which covers various phases of the disaster management process, consists of 52 questions, with a final scale of 34 items centered on three main factors: post-disaster management, knowledge-related skills ('knowing'), and practical skills ('doing'). The fidelity of the DPET-PT was checked by analyzing its internal consistency using Cronbach's alpha coefficient, which was initially determined using all 45 items, resulting in a total alpha of 0.949. However, 11 items had to be removed because they had Cronbach's alpha values higher than the overall value and, in some cases, correlation values without the item lower than 0.20. As a result, the DPET-PT scale was left with 34 items.
The nurses in this study perceive their average level of preparedness for disaster situations as low/weak. However, the factor in which nurses are best prepared is competencies related to 'knowing.' In this pilot study, the training needs most pointed out by the nurses are the role (competencies) of the nurse in a disaster situation and the resources in the community, reference contacts, health departments, emergency contacts, chain of command, and community shelters.”
Comments 4: Please rewrite the results in paragraphs and avoid the dot points. You may provide little explanations for tables including demographics, findings and inferential analysis.
Response 4: Agree. We rewrite the demographics result in line 249-259, inferential analysis in line 314-334 and findings in line: 297-312; 349-359
“The characterization of the sample of 230 nurses participating in the study reveals a specific demographic and professional profile (supplement table 1-4): most of the sample is female, representing 75.2% (173 nurses), the predominant age group is between 33 and 43 years old, corresponding to 41.7% (96 nurses), more than half of the participants were married, totalling 56.1% (129 nurses), the vast majority have a bachelor's degree (88.3%, 203 nurses), while 11.7% (27 nurses) have a master's degree, 58.3% (134 nurses) work in hospitals, the typical shift pattern is "Rolleman," practiced by 53.9% (124 nurses), 30% (33 nurses) have a specialization in Medical-Surgical Nursing. Most works on São Miguel Island (71.3%, 164 nurses) and 91.3% (210 nurses) are on islands with a hospital, 82.6% (190 nurses) have open-ended public service contracts, 81% (187 nurses) work 35 hours a week and the majority, 58.7% (135 nurses), are generalists.”
Inferential analysis
“The study highlights significant differences in disaster preparedness between male and female nurses across various dimensions, all showing a p-value of less than 0.01. Regarding disaster knowledge, men had a higher average score of 3.65 (±0.89) than women, who scored 3.28 (±0.78), indicating that men possess more extensive theoretical knowledge. Similarly, in disaster competence, men scored significantly higher, with an average of 3.22 (±1.09), while women recorded an average of 2.45 (±1.06), suggesting that men feel more competent in their disaster response capabilities.
Regarding family preparedness for disasters, men reported an average score of 2.83 (±1.64), whereas women scored 2.30 (±1.40), indicating that men perceive themselves as better prepared to assist their families during such situations. In specific knowledge of disasters, men achieved an average score of 3.13 (±1.31) compared to women’s 2.62 (±1.11), reflecting more excellent detailed knowledge among men. Furthermore, during patient management in response to disasters, men showed an average score of 3.04 (±1.07), while women scored 2.43 (±0.94), indicating that men feel more capable of managing patient care during disaster responses.
When examining knowledge about recovery, men scored 3.35 (±1.32) compared to women’s 2.79 (±1.42), signifying a greater understanding of recovery processes among men. However, in recovery management, there were no statistically significant differences between genders, with men averaging 2.41 (±1.18) and women 2.10 (±1.04) and a p-value of 0.061. Overall, the analysis indicates that, except for recovery management, male nurses generally report higher levels of knowledge and competence in disaster preparedness when compared to their female counterparts.”
Differences between professional roles: line 349-359, 370-385
“The study revealed significant differences in specific dimensions of disaster preparedness related to the nurses' professional roles. Nurses in managerial or coordinating positions demonstrated higher scores in disaster knowledge, with Nurses in Charge averaging 3.93 (±0.98) and Nurses in a Coordination Role scoring 4.15 (±0.95), both outperforming Generalist Nurses, who had an average of 3.21 (±0.70), and Specialists, who scored 3.45 (±0.88), with the coordination role particularly standing out. Regarding competence in disaster management, nurses in a coordinating role achieved significantly higher scores, averaging 3.41 (±1.41), compared to general nurses, who averaged 2.53 (±1.03). Furthermore, nurses in a coordinating role scored 3.85 (±1.63) in recovery knowledge, showing notably more excellent knowledge than specialist nurses, who had an average score of 2.66 (±1.41).”
Comments 5: Please rewrite the discussion in paragraphs instead of dot points and critically discuss the main findings with other literature of climate change.
Response 5: Agree. We integrate the Keyword “Climate Change” in line 36 and remove disaster, disaster preparedness and disaster management stay as keywords. Addressing the broader implications of climate change is important so we can better prepare and adapt our health systems to meet these emerging challenges.
We have restructured the discussion into paragraphs and removed the bullet points. Additionally, we have incorporated four new pieces of literature that address the impact of climate change on health.
We restructured the conclusions aligned with the discussion and the climate changes.
Line 595-608
“The world faces growing threats due to increased natural disasters, extreme meteorological phenomena, changes in agricultural and fishing areas, loss of biodiversity, and flooding of cities and even entire countries. The fight against climate change requires collective action and a coordinated response from the international community (25). In the context of the climate agenda, the involvement of nurses in the defense of causes and policies for the benefit of public health is essential. Nurses' leadership is crucial in tackling health and social crises, and it is vital to understand the structural and social determinants of communities (26). Additionally, equipping nurses with the skills to manage disaster situations—emphasizing prevention and well-being—is essential for reducing vulnerability to climate risks, enhancing resilience to extreme weather events, and improving emergency response capabilities. In this study, 70% of participants reported a low capacity to respond, highlighting an urgent need to address this shortfall. These findings align with previous surveys indicating a global inadequacy in nurses' disaster preparedness.”
Line 610-614
“The adverse effects of climate change represent a significant challenge for humanity worldwide, contributing to the global disease and premature mortality burden (29). Understanding the impact of climate change, mainly through the lens of natural disasters and effective adaptation strategies, is crucial for enhancing resilience against its harmful consequences. “
Line 615 -629
“Previous experience is crucial for knowing and preparing nurses for disaster response. The study revealed that only 14.3% of nurses had engaged in an absolute disaster; of these, only 42.6% felt prepared. Active involvement in disasters boosts confidence and allows nurses to apply theoretical knowledge practically. Previous experience makes it possible to identify a community's vulnerabilities and consequences. Hossain et al. (30), in their study, identified hunger (83%) and physical injuries (56.8%) as the most significant complications of climate changes. Difficulties in obtaining first-aid medicines to deal with sudden health problems or access to health institutions that have been closed due to the storms contribute to the community's increased vulnerability. Increased exposure and training lead to improved readiness, as supported by Ziapour and Darabi (4), who found that emergency nurses with both experience and training scored higher in preparedness. Nurses with experience and specific knowledge of the adverse effects of climate change can contribute positively by identifying adaptation strategies to increase access to food quickly and adequately, reduce traumatic events, or adapt resources for an adequate response.”
Line 630-634
“Previous disaster situations enhance understanding of roles and expected behaviour. Practical knowledge in real scenarios is often more valuable than years of experience alone. While nurses understand triage, comprehensive training and disaster management simulations are needed. Mastering first aid and triage is crucial for nurses to provide immediate care and lifesaving interventions in disasters, as noted by Santos and Rabiais (5).”
Line 643-648
“Hassan et al. (31) emphasize the need for health professionals to be more aware of climate change and its negative impact on a community's health. Active involvement in disaster enhances nurses' clinical judgment and preparedness for disaster events, as practical experience develops skills beyond theoretical training. The study highlights a critical need for better disaster preparedness among nurses. Barriers to effective response include a lack of family emergency plans, impacting nurses' focus on disaster management.”
Line 651-653
“This situation can be circumvented if family members and neighbors support one another when faced with this problems (30).”
Line 654-663
“Understanding the public's vulnerability to the health impacts of climate change is crucial for timely preparation and early prevention (30). Azorean nurses are aware of potential disaster risks and acknowledge their limitations in knowledge and competence. However, they feel inadequately prepared to manage these situations and express a strong need for additional training. Effective planning addresses disasters, mitigates risks, ensures safety, and maintains operational continuity. Low literacy and low wealth levels are associated with greater vulnerability (30, 31). Health institutions should create proactive risk management plans that involve healthcare professionals in their development. Engaging nurses in disaster planning enhances risk awareness and promotes early readiness, contributing to positive outcomes.”
Line 664-665
“Nurses who are involved in drawing up and implementing disaster plans are essential for dealing with health and social crises (26, 30-33).”
Line 671-672
“comprehensive knowledge of the health impacts of climate change for”
Line 681
“The results of this study”
Line 684-686
“Prioritize education to recognize the signs and symptoms of climate change-related diseases, and improve competencies in adaptation, mitigation, and resource mobilization at all levels (31).”
Line 690-694
“One significant impact of climate change is the 40% increase in emergency room visits due to trauma (adult and pediatric), emergencies due to carbon monoxide poisoning, skin infections, respiratory diseases, medical illnesses, and psychotic outbreaks due to mental health decompensation (32), which means that specialized nurses are needed for the different age groups, as well as knowledge of the different areas of activity.”
Line 713-714
“and involve the media as a means of raising awareness and disseminating information to the community (31).”
Line 720-723
“the region's isolation from the mainland is a reality, which can lead to a reduction in food supplies, technological resources that are unable to function and difficulty in accessing healthcare, placing the Azores in a situation of increased vulnerability.”
Line 731-733
“Training should include face-to-face activities, such as simulations and crisis response exercises, which can improve the effectiveness of disaster management education (34).”
Line 743-747
“The need for leadership and collaboration at regional, national, and international levels is important for smaller countries with limited resources to combine efforts and be more effective and efficient in disaster management. Disruption of access to healthcare due to damage to medical facilities, power, transport, communications, or other infrastructure due to extreme weather-related disasters can negatively affect people (31).”
Line 774-775
“of the negative impacts of climate change, with a preventive and adaptative perspective”
Line 795-796
“reduce the negative impacts of the climate changes,”
Line 814-820
“The study underscores the critical need for enhanced disaster preparedness among nurses, particularly in disaster-prone regions such as the Azores. With the increasing impacts of climate change, it is urgent to recognize the importance of disaster training and implement essential procedures to ensure effective responses. Continuous education and training programs that seamlessly integrate theory and practice in disaster scenarios are vital for strengthening nurses' capacity to act decisively. Training should encompass family safety planning, which can help reduce anxiety and increase focus during disasters.”
Line 821-830
“Moreover, it is essential to tailor training to regional needs, especially in areas lacking hospital resources, to promote ongoing knowledge regarding disaster response plans and the specific challenges posed by climate change.
Additionally, healthcare institutions should prioritize the development of leadership skills among nurses, enabling them to guide their teams effectively during crises.
This proactive approach will enable nurses to fulfil pivotal roles in disaster response and community resilience, ultimately benefiting both healthcare professionals and the communities they serve in times of crisis exacerbated by climate change. By fostering leadership, improving knowledge, and enhancing community resilience, we can create a more robust healthcare system prepared to face future challenges.”

Reviewer 2 Report
Comments and Suggestions for Authors
Thank you for your hard work on this article.
Line 15 Climate increases - this needs further explanation - for example Extreme climate or climate change
30 to 31 - overlong sentences and lack of punctuation
Introduction 41 to 45 you need to rewrite this to cover the two mains strategies as being mitigation and preparedness/ The first paragraph will be your road map to the rest of this section. Does prevention play a role in your disaster management
Line 52/53 recent studies show that there is little research...
line 57 rewrite for clarity
Line 60 I would include advanced critical care skills, resource allocation, care of both physical and mental health needs
Line 52/53 recent studies show that there is little research...
Line 67 what is vision 2030 - explain for your reader
Line 93 - further information needed - reader may not understand what disclosed by... means
Line 103 this approach aimed
Line 108 Tichy et. al.
Good approach to ethics! Thank you
Note that the stages of Disaster Management are usual considered to be Mitigation, Preparedness, Response, and Recovery. https://training.fema.gov/emiweb/downloads/is111_unit%204.pdf. Please set up your data to demonstrate this order.
Table 2 looked at islands with a hospital however you stated it would look at both or am I missing something?
line468 use a more academic term to replace tiny
You study is important but it needs work on rewriting it. Remember that your readers may not be familiar with the Azores and the working environments etc so it is necessary to explain these.
It would be nice to see graphs of your tables for people who are visual learners.
These comments do not detract from the work you have done on this study, they are intended to help you with publication and future work. With some changes this will add to the body of knowledge on disaster management.
You will need to review and rewrite sections of your paper. Remember your readers and they may not be well versed in your geographic area or disaster preparedness needs
Author Response
Comments 1: Line 15 Climate increases - this needs further explanation - for example Extreme climate or climate change
Response 1: Agree. We change for “Climate changes increase”, line 15
Comments 2: 30 to 31 - overlong sentences and lack of punctuation
Response 2: Agree. We have changed the sentence to emphasize de nurse contributions on climate change. The change can be found at page 1, line 30 to 35.
“Despite increased knowledge and awareness of climate change and its global impact, there is a need for more meaningful nursing intervention to minimize the impact of climate change on the community. This study highlights that nurses have an in-depth knowledge of communities' needs, which puts them in a unique position to influence disaster preparation and response. By developing these strategies, nurses contribute significantly to communities' resilience in climate change, improving society's ability to respond and adapt to these challenges.”
Comments 3: Introduction 41 to 45 you need to rewrite this to cover the two mains strategies as being mitigation and preparedness/ The first paragraph will be your road map to the rest of this section. Does prevention play a role in your disaster management
Response 3: Agree. We have changed the sentence to define the two main strategies and the importance of the preparation of communities to prevent vulnerabilities of the community.
The change can be found at page 2, paragraph 1, line 42-53.
“The effects of climate change are evident; the world is threatened by increased natural disasters and extreme weather events, changes in traditional agricultural and fishing areas, loss of biodiversity, and flooding of entire cities, which are increasingly threatening human life, natural systems, and global economies. In Portugal, the Basic Climate Law was drawn up to tackle the problem of climate change, and there are two lines of action: mitigation and adaptation. Mitigation reduce greenhouse gas emissions into the atmosphere, and adaptation minimize the adverse effects of climate change on biophysical and socio-economic systems. By preparing the community for adverse events resulting from natural disasters caused by climate change, nurses prevent vulnerability in communities and allow them to adapt to a new reality with low biophysical and socio-economic impact, emphasizing the importance of their work in preparing the community.”
Comments 4: Line 52/53 recent studies show that there is little research...
Response 4: Agree. We changed to authors names, line 60-61.
Comments 5: line 57 rewrite for clarity
Response 5: Agree. We introduce the climatic characterisation of the Azores for a better understanding of the problem. Paragraph 2, line 54-60.
“The general climate conditions in the Azores are determined by their geographical location in the context of global atmospheric and oceanic circulation and by the effect of the vast body of water surrounding them. The climate of the Azores is temperate maritime, reflected by the low-temperature range, high rainfall, relative humidity, and persistent winds. These are often affected by the passage of tropical cyclones or tropical storms derived from these and usually result in the worst storms to which the archipelago is subjected.”
Comments 6: Line 60 I would include advanced critical care skills, resource allocation, care of both physical and mental health needs
Response 6: Agree. We revised the sentence. Paragraph 4, line 70-74.
“In the Azores, nurses must be prepared for disaster situations to develop advanced skills in dealing with people in critical situations, such as triage, infection control, communication, allocating human and material resources, meeting physical and mental health needs, and leadership to ensure adequate response and minimize the impact of the disasters.”
Comments 7: Line 67 what is vision 2030 - explain for your reader
Response 7: Agree. We revised the sentence. Paragraph 5. Line 78-84.
“In 2019, the Government of the Azores developed the Regional Climate Change Programme. One of its strategic objectives is to stablish climate scenarios and projections up to 2030 and define program climate change mitigation and adaptation measures. Nurses have a unique opportunity to develop and implement education and preparedness strategies as a form of a disaster prevention and a research agenda that responds to current concerns that negatively affect health, including research into the impact of climate change on individuals, families, and communities worldwide.”
Comments 8: Line 93 - further information needed - reader may not understand what disclosed by... means
Response 8: Agree. We explain in line 106-109.
“The study sample included 230 nurses from the Azores islands, except Corvo. The questionnaire was emailed to all nurses registered with the Nursing Council of the Azores and publicized in closed groups on nursing social networks; however, there is no record of participation from the nurses of the island of Corvo.”
Comments 9: Line 103 this approach aimed
Response 9: Agree. Changed to aimed, now line 119
Comments 10: Line 108
Response 10: Agree. Changed to Tichy et. al., now line 124
Comments 11: Note that the stages of Disaster Management are usual considered to be Mitigation, Preparedness, Response, and Recovery. https://training.fema.gov/emiweb/downloads/is111_unit%204.pdf. Please set up your data to demonstrate this order.
Response 11: Agree. We change Mitigation and Relief Phase to first, as can be seen in line 295.
Comments 12: Table 2 looked at islands with a hospital however you stated it would look at both or am I missing something?
Response 12: Agree. Thank you for point this out. The table analyses islands with and without a hospital, separating "Yes" from "No”." Yes corresponds to n=210 (with Hospital), and No (Without Hospital) corresponds to n=20. We have added the missing analysis of the table, in line 406-427.
“In terms of knowledge of disasters, nurses on islands without hospitals had an average score of 4.02 (±0.91), compared to 3.31 (±0.79) for those on islands with hospitals. The t-value was 3.771, and the p-value was 0.000, indicating a statistically significant difference. For disaster competence, the average score for nurses in non-hospital islands was 3.43 (±0.92), whereas those in hospital islands scored 2.57 (±1.10). The t-value here was 3.390, with a p-value of 0.001, reflecting a significant difference. When preparing families for disaster, nurses in non-hospital settings averaged 3.30 (±1.94), while those in hospital settings scored 2.35 (±1.40). The t-value was 2.147, and the p-value of 0.044 suggests a significant difference. In specific knowledge of catastrophes, nurses without hospitals had an average score of 3.77 (±1.25), compared to 2.65 (±1.13) for those with hospitals. This resulted in a t-value of 4.174 and a p-value of 0.000, indicating strong statistical significance. For patient management during the response, non-hospital nurses averaged 3.21 (±0.94), whereas hospital-based nurses scored 2.52 (±1.00). This yielded a t-value of 2.969 and a p-value of 0.003, demonstrating significance. In knowledge about recovery, nurses in non-hospital settings had an average of 3.80 (±1.28) compared to 2.84 (±1.40) for those in hospitals, with a t-value of 2.948 and a p-value of 0.004, indicating a significant difference. Finally, regarding recovery management, nurses in non-hospital islands averaged 2.86 (±1.19), while those in hospitals scored 2.12 (±1.05). The t-value was 2.983, and the p-value of 0.003 shows a statistically significant difference between the two groups.”
Comments 13: line468 use a more academic term to replace tiny
Response 13: Agree. We change the word tiny to small. This change is in the line 539
Comments 14: It would be nice to see graphs of your tables for people who are visual learners.
Response 14: Agree. We change the table1 to figure 1 and 2 and the table 3 with figure 3.

Round 2
Reviewer 1 Report
Comments and Suggestions for Authors
Thank you for the revisions. I have enquiry regarding the instrument, why did you included items for both competencies and knowledge? Are they different, so what are the difference between them. Also the instrument of DEPT, are used for disaster preparedness in general, how did you modify it to fit for environmental disaster, what did you include?
Author Response
Comments 1: I have enquiry regarding the instrument, why did you included items for both competencies and knowledge? Are they different, so what are the difference between them.
Response 1: Thank you for pointing this out. We include the differences in lines 138-146
“Disaster knowledge and competencies complement each other but focus on different elements. Knowledge focuses on concepts and information about disasters and includes understanding the types of disaster, their causes and effects, risk factors, historical events, risk assessment and emergency planning. Competencies refer to practical skills needed to respond to disasters, such as triage, first aid, infection control, communication, and resource allocation. Both are essential for effective preparedness and response, increases confidence for autonomous decisions in unpredictable scenarios, reducing the negative impact of disasters and improve patient outcomes.”
Comments 2: Also, the instrument of DEPT, are used for disaster preparedness in general, how did you modify it to fit for environmental disaster, what did you include?
Response 2: Thank you for pointing this out. In lines 604 to 613, we explain the importance of nurses' involvement in defending causes and policies that benefit public health. By identifying the results of climate change in the community and the need for nurses' disaster preparation, we can reduce the community's vulnerability to climate risks, enhance resilience to extreme weather events, and improve emergency response. Also, in lines 675 to 678, we state that health institutions should establish comprehensive knowledge of the health impacts of climate change for disaster plans and provide sufficient training to nurses to ensure effective interventions during and after disasters.
Prioritize education to recognize the signs and symptoms of climate change-related diseases can improve competencies in adaptation, mitigation, and resource mobilization at all levels (lines 675-678; 694-698; 723-726; 746-750).
Reviewer 2 Report
Comments and Suggestions for Authors
There are a few grammer and syntax errors that you will pick ou in editing otherwisse well done on responding and responding in a timely manner.
You will add to the body of knowledge on disaster response!
Author Response
Thank you for your feedback. I appreciate your guidance and will review the grammar and syntax errors during editing.
I'm grateful for the opportunity to contribute to the body of knowledge on disaster response.
Best regards.